# Testing Road Vehicle User Interfaces Concerning the Driver's Cognitive Load

**Viktor Nagy** [1], **Gábor Kovács** [1], **Péter Földesi** [1,2], **Dmytro Kurhan** [3], **Mykola Sysyn** [4], **Szabolcs Szalai** [1,*] **and Szabolcs Fischer** [1,*]

[1]   Central Campus Győr, Széchenyi István University, Hungary, Central Campus Győr, H-9026 Győr, Hungary; nviktor@sze.hu (V.N.); gkovacs@sze.hu (G.K.); foldesi@sze.hu (P.F.)
[2]   Eötvös Loránd Research Network, Piarista u. 4, H-1052 Budapest, Hungary
[3]   Department of Transport Infrastructure, Ukrainian State University of Science and Technologies, UA-49005 Dnipro, Ukraine; d.m.kurhan@ust.edu.ua
[4]   Department of Planning and Design of Railway Infrastructure, Technical University Dresden, D-01069 Dresden, Germany; mykola.sysyn@tu-dresden.de
[*]   Correspondence: szalaisz@sze.hu (S.S.); fischersz@sze.hu (S.F.); Tel.: +36-(96)-613-544 (S.F.)

**Abstract:** This paper investigates the usability of touch screens used in mass production road vehicles. Our goal is to provide a detailed comparison of conventional physical buttons and capacitive touch screens taking the human factor into account. The pilot test focuses on a specific Non-driving Related Task (NDRT): the control of the on-board climate system using a touch screen panel versus rotating knobs and push buttons. Psychological parameters, functionality, usability and, the ergonomics of In-Vehicle Information Systems (IVIS) were evaluated using a specific questionnaire, a system usability scale (SUS), workload assessment (NASA-TLX), and a physiological sensor system. The measurements are based on a wearable eye-tracker that provides fixation points of the driver's gaze in order to detect distraction. The closed road used for the naturalistic driving study was provided by the ZalaZONE Test Track, Zalaegerszeg, Hungary. Objective and subjective results of the pilot study indicate that the control of touch screen panels causes higher visual, manual, and cognitive distraction than the use of physical buttons. The statistical analysis demonstrated that conventional techniques need to be complemented in order to better represent human behavior differences.

**Keywords:** road safety; cognitive load; human factor; ergonomics; driver attention; driver distraction

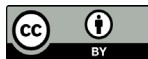

## 1. Introduction

The transformation of individual transportation due to the advancement of new automotive technologies is inevitable. Automated, semi-autonomous, and autonomous vehicles are spreading; this facilitates the change of user interface (UI) development. The automation level of mass production cars is on SAE level 2, "Partial Driving Automation"; this means that Advanced Driving Assist Systems (ADAS) can control vehicle lateral and longitudinal motion with the expectation that the driver completes the object and event detection and response [1]. Multiple controllable functions and the increased number of Non-driving Related Tasks (NDRTs) require more competence and attention for the same driving tasks and increased cognitive load [2].

Vehicles are still controlled manually by the driver who is the operator in the context of Human-Computer Interaction (HCI). This raises the question: if present Human-Machine Interfaces (HMI) are able to manage proper communication and support the required safety level, what is promised by ADAS? Today, HMIs and HCI concepts are built

for manually controlled road vehicles. The cognitive demand of on-board interfaces can cause more manual and visual distractions than are acceptable in a moving vehicle [3].

Automated systems (ADAS) are not perfect; they can make incorrect decisions and errors in some situations. As humans are the only flexible and adaptable part of the system, only they can prevent these errors by modifying the processes of the system if necessary. Situational awareness of the human driver is critical in automated or aided driving [4].

Based on traffic injuries data, there has been no significant decline in global road traffic death rates between 2010 and 2016 [5]. Only a slight improvement is seen when adopting new road safety systems and safer vehicles. In low-income countries, traffic death rates have increased in the last decade. A model study using the raw traffic accident data of Montenegro showed that with an increase in the number of motor vehicles, there is a decrease in the number of deaths in traffic accidents (thanks to preventive engineering and actions); however, with an increase in the number of motor vehicles, there is also an increase in the number of traffic accidents [6].

Before fully automated road vehicles become widespread, traditional Driver-Vehicle communication processes and technologies should be examined. Simple NDRTs are the source of driver distraction, and the severity of this distraction depends on the type and ergonomics of the User Interface (UI). The primary objective of this research was to investigate visual demand and driver distraction. More studies have used the occlusion technique to evaluate IVIS displays. The occlusion procedure simulates the visual demands of driving in a stationary setting using a set of glasses with lenses that are either transparent or opaque. When participants' vision is blocked by the opaque lens, it is like looking at the road, not the dashboard. The complex 15 s long tasks were chunked into smaller units to make the measurement method more manageable [7,8]. This technique is used for assessment of driver behavior when drivers are completing basic NDRTs [9]. The goal of the study was to validate the results in a low-fidelity simulator and propose the acceptance criteria of NHTSA's Driver Distraction Guidelines [10].

Another approach in distraction detection is to use different types of eye tracking methods to obtain more satisfying glance behavior results. It was reported that there is a correlation between fixation rate and accident rate and road complexity level [11]. Further research has found that different NDRTs have different effects on driver behavior. Auditory and audiovisual stimuli generate faster reactions when compared to visual information [12]. A naturalistic driving study examined three in-vehicle input and control types (a physical dial, pressure-based input on a touch surface, touch input on a touchscreen) to investigate their effects [13]. An in-depth usability analysis was performed for a touch button interface that examined touch button size, color, shape, and other design combinations [14]. Furthermore, more recent input modalities (touch, speech, and gesture based) were investigated in a simulator experiment where gaze data and off-the road glances showed driving and visual performance differences [15].

The primary goal of our research study was to find an appropriate methodology to investigate and compare traditional and newly developed system elements of HMI in cars. A comparison of automotive UIs was performed using a physiological measurement system that detects human drivers' distraction. A modular test system was designed earlier, and some elements of that setup were used for this pilot study [16].

### 1.1. Human Machine Interface

HMIs were developed to ensure proper interaction between the vehicle and driver. HMIs in vehicles (road and railway) consist of output and input channels. The output channels provide information about the system status to the driver (e.g., through displays and audio signals), or energy consumption and energy saving capabilities using an advanced monitoring and assistant system for train operators [17]. The input channels receive the driver's intention to input information (e.g., via buttons, steering wheel, pedals) [18]. Typical production vehicle HMI input interfaces are dedicated tactile switches

(physical buttons, rotary knobs, push levers) combined with menu based on-screen projected control panels (mono- or multi-colored touch screens, or center console mounted rotary controllers). New automotive developments are implementing speech recognition, handwriting recognition, and gesture recognition for NDRTs [19]. The increasing level of instrumentation and motivation of car manufacturers adding advanced technologies to premium vehicles led to increased sophistication of IVIS [20]. These multi-colored touchscreens, that have quickly become widespread, are expected to ensure higher usability and performance, especially with capacitive technology and high resolution [21]. Screen size and other Graphical User Interface (GUI) design characteristics, such as button size, button spacing, button shape, button color, and visual or haptic feedback, also affect touchscreen efficiency, alongside input performance, usability, and a driver's cognitive load [14,22,23]. Physical parameters (screen size, button quantity) and ergonomics of IVIS are still important. Many dashboards and interface panel design options are used, which have different safety awareness and usability features [24]. Functionally, IVIS is responsible mainly for basic vehicle settings, such as infotainment (e.g., music, phone calls), integration of nomadic devices (e.g., smartphones), advanced navigation, and other comfort features (e.g., climate control, seat heating, external view cameras).

### 1.2. Driving Tasks and Non-Driving Tasks

The complexity of driving a road vehicle involves more than operating pedals and turning a steering wheel. Three task types can be defined in relation to driving, as follows [25–27]:

1. **Primary Driving Task:** Maneuvering the car, hazard detection (e.g., controlling speed, checking the distance to other road users), and hierarchically cascaded tasks (navigation, guidance, stabilization).

2. **Secondary Driving Task:** Functions that increase safety (e.g., turning signal, windshield wipers).

3. **Tertiary Driving Task** (NDRT): All functions regarding entertainment, comfort, and information systems.

### 1.3. Driver Distraction

In the literature, driver distraction is defined as "the diversion of attention away from activities critical for safe driving toward a competing activity" [28]. Driver distraction is part of a broader definition of driver inattention. Driver inattention can be distinguished as Driver Diverted Attention, Driver Restricted Attention, Driver Misprioritized Attention, Driver Neglected Attention, and Driver Cursory Attention. Driver distraction is a synonym of Driver Diverted Attention (DDA), that includes Driving-Related and Non-Driving-Related attention or inattention. DDA is a distraction that restricts activities necessary for safe driving [29]. Driver inattention estimation can be analyzed with glance behavior analysis using eye tracking and proper algorithms, but drivers still have visual spare capacity or off-target glances [30]. The US-based National Highway Traffic Safety Administration (NHTSA) differentiates four types of distractions: visual, auditory, biomechanical (manual or physical), and cognitive. Detailed definitions are as follows:

- **Visual Distraction**: the driver's field of vision is blocked by objects that prevent the perception of the road and its surroundings; loss of visual "awareness" that hinders the driver's ability to recognize hazards in the road environment; driver does not focus attention on the road but on another visual target (e.g., on-board navigation system) [31].

- **Auditory Distraction**: driver temporarily or continually focuses attention on sounds or auditory signals rather than on the road environment. The source of auditory distraction can be the radio or cell phone calling, which is not significantly risky [32] but still has a distracting effect through higher cognitive demand [33].



- **Manual** (**Biomechanical or Physical**) **Distraction:** if drivers remove one or both hands from the steering wheel to manipulate an object instead of focusing on driving (steering or changing gears), this can cause a lower reaction time and less capacity to avoid dangers [34].
- **Cognitive Distraction**: cognitive distraction includes thoughts that restrict drivers' attention focusing on driving tasks. This is the "Look at but not see" problem, which is when drivers are unable to navigate through the road network safely [34]. It is often caused by external factors (e.g., using mobile phone, talking to a passenger), heavy cognitive load (e.g., operating in-vehicle devices), bad physical conditions, or fatigue.

### 1.4. Cognitive Load

Cognitive load can induce a type of error when cognitively loaded drivers reach the inability of flexibly to adapt to novel or unusual driving situations. Cognitive load selectively impairs (non-automatized) driving performance, which relies on cognitive control but leaves automatized tasks unaffected. However, it is possible that cognitive load has other effects that play a key role in the genesis of severe crashes. The findings from naturalistic driving studies investigating the relation between cognitively loading tasks and crash risk are not simple. It seems important to conduct more detailed analyses of naturalistic crashes involving cognitively loaded tasks in order to understand its impact on general road safety [35].

### 1.5. Observing Cognitive Load

In recent years, based on the results of research on the subject, it has been proven that the frequency and duration of fixation, the significant change of pupil diameter from the normal size, quantity of saccades and microsaccades, and the frequency and duration of blinking are closely related to cognitive load [36,37]. The continuous focusing of the gaze on a single place is called fixation. Fixation is determined by the average of the x-y coordinates of the gaze position. The Point Of Regard (POR), that is the point in space that the gaze observes for a given time period, must remain within a certain area for a minimum period of time in order for it to be called a fixation [38]. Using driving simulations with controlled environmental parameters (e.g., ambient lights), it was shown that average fixation duration is negatively correlated with cognitive load. With the help of simulation tests, it was shown that the number of fixations increases to a similar extent as the cognitive load; the average duration of fixations during the process of visual processing in a road traffic environment is between 90 and 300 m [39,40]. Other studies where more variables were disclosed by using driving scene videos have shown that fixation behavior and patterns correlate with driving experience [11,41].

Pupil size reflects autonomic involuntary activity; this is often associated with cognitive load. Pupillometry is a non-invasive procedure that examines changes in pupil size and movement with the aim of obtaining information about brain activity. Numerous studies have confirmed that pupil size shows a positive correlation with the level of cognitive load. The diameter of the pupil is generally 4 mm, but this can vary between 1 mm and 9 mm due to various factors. The pupil has been proven to react to light, pain, feelings, and cognitive load, among other factors. Cognitive pupillometry experiments usually use stimuli from several conditions. In other studies, average pupil size provided clear feedback on multitasking while driving. After performing tasks on IVIS of varying difficulty, it was observed that the average pupil size increased significantly [40,42]. Verbal tasks, spatial imagery [43], and mental tasks resulted in an increase in pupil size, indicating more mental effort and gaze concentration [44].

A saccade, or saccadic movement, is a jump-like, simultaneous fast tandem movement of both eyes that occurs during fixation of the gaze from one object to another, in the same direction as the movement between the two fixation points. During this

phenomenon, the subject does not perceive visual information. The length of saccadic movements is usually 30–80 m and their amplitude is 0.3° [45]. This amplitude may vary depending on the task. When it comes to cognitive load, the examination of microsaccades cannot be neglected either, as it has also shown a correlation with cognitive load. Microsaccades are the very small gaze changes when the goal is not to change the gaze from one point to another, but to keep the gaze in a specific position, i.e., fixation. The frequency of microsaccades decreases during the execution of a mental task. In contrast, during a test performed in a driving position, it was shown that while a driver was performing a secondary task, the frequency of saccades increased significantly [46].

Blinking is the rapid and semi-involuntary movement of the eyelids. The average duration of a blink is 0.1 and 0.4 s. Blink frequency is measured in blinks/second. This value can be easily influenced by changing environmental conditions or physical activity. Nevertheless, the duration of the blinks can effectively measure cognitive load. As a result of visual load, frequent short blinks are typical, while during the execution of a difficult task, the frequency of blinking displayed a continuous tendency to slow down. In a driving experiment, the eye movement tracking camera showed an increase in blinking frequency parallel to the performance of the secondary task performed while driving [38,47].

### 1.6. Muscle Memory

Muscle memory or motor learning is a type of procedural memory that involves the memorization of a given motor task by repetition, supplemented by the process of motor learning [48] In theory, muscle memory allows the driver to move their hands from the steering wheel directly to the controls built into the vehicle without shifting their gaze. The development of muscle memory is easier when using traditional, physical controls (e.g., buttons, knobs), as the driver knows roughly where the button is, and can then use tactile information to help distinguish one from another by touching the buttons and getting physical feedback. This is not possible with touchscreen displays, which lack tactile feedback (a haptic solution could be introduced in which the touch screens provides the feel of pressing a button when touched). This makes it difficult for the driver to develop muscle memory, or if some muscle memory is developed, it will only direct him to a particular area of the screen where the icon (smoothed button) in question is located [49].

### 1.7. Present Study

Based on the literature review, driver distraction caused by NDRT using IVIS is a current problem even if conventional UI elements are investigated. Increasing numbers of multimodal systems are integrated in current series production cars; however, a detailed comparison is needed to evaluate them. A narrow gap was found in the research in which there is a direct comparison of conventional (tactile) and touch interface, and both together as an integrated system of cars that are currently mass-produced.

Our focus was on monitoring and identifying visual, manual, and cognitive distractions using an eye-tracking system and psychological questionnaires. The objective of the research was to examine IVIS with physical buttons compared to touch screen interfaces. The comparison analysis used conventional statistical tests and, as a new approach, Type-2 fuzzy sets were added for better representation of results.

Driver inattention, specifically distraction, is situation dependent and affected by external confounding variables [50,51]. To reduce or minimize the impact of these factors, a straight road and simple short tasks were selected for our naturalistic driving study. The tertiary climate control tasks that were incorporated into the study were a focused 1.42–6.29 s long process. While the tasks were completed, driver attention was focused on them. The participants' low-load primary task was to drive with constant speed, keeping in the lane while no other distractions arose.

Our hypothesis is that one of the interface designs causes a higher, or significantly higher, distraction that results in a higher cognitive load for the driver.

## 2. Experiment

The experiment investigated the driver's behavior, cognitive load, and level of distraction from the point of view of the distracting effect of the IVIS examined with regard to traffic safety. Visual, manual, and cognitive distraction was measured using different test methods, including psychological tests, a questionnaire, and an eye-tracking monitoring system.

The pilot test was carried out on 16 participants (N = 16) as volunteers (they were not compensated). They did not wear corrective glasses and did not have an eye-related illness or surgery before the exam. The participants had different lengths of driving experience. The three female and thirteen male participants were 20 to 44 years old (Mean 26.9 years, Standard Deviation 16.4). All drove regularly and were familiar with road vehicles equipped with standard IVIS in Volkswagen Group cars produced from early as 2012. The naturalistic driving test took place on a closed track at ZalaZONE Test Center (Zalaegerszeg, Hungary), on the "High-Speed Handling Course" [52]. The track has the following physical characteristics: a length of 2000 m, a width of 12 m, a soft gravel/asphalt run-off-area, an 80 cm basalt base/foundation, and a 450 m long straight section. The test vehicle was a fully equipped Volkswagen e-Golf from the year 2020 (Figure 1).

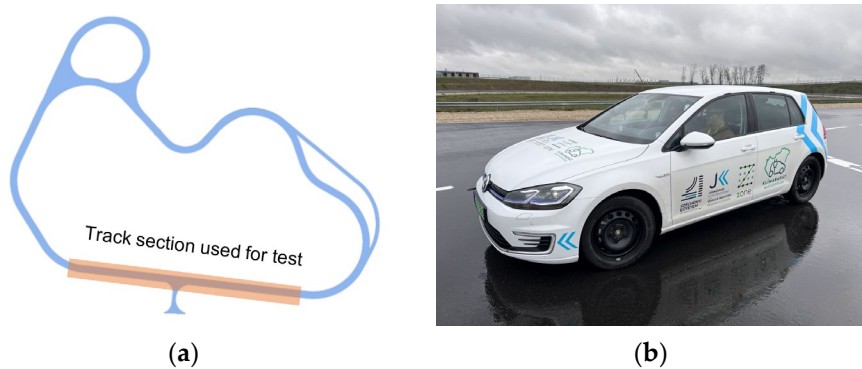

| (**a**) | (**b**) |

**Figure 1.** (**a**) 450 m long straight section of ZalaZONE "High-Speed Handling Course"; (**b**) Test vehicle: VW e-Golf.

### A. Apparatus

Visual, manual, and cognitive distraction was detected using a wearable eye-tracking device supported by high-definition cameras (Figure 2). The head-mounted eye-tracker device (Pupil Labs Core) was used for gaze detection and monitoring pupillometry characteristics. The binocular glasses of Pupil Labs were chosen because they are more accurate than other similar devices (e.g., SMI ETG 2.6, Tobii Pro Glasses 2) [53]. The glasses were equipped with 2 infrared (IR) eye cameras (120 Hz@400 × 400px) and one RGB world-view camera (30 Hz@1080p) with a 139° × 83° wide-angle lens. The extensible, open-source platform mobile eye tracking system contains recording and player/analytics software with GUI to visualize video and gaze data [54]. The recording was managed using Pupil Capture software installed on a high-performance mobile workstation PC (11th Gen Intel(R) Core (TM) i7–11800H, 32 DDR4 RAM, NVIDIA GeForce RTX 3050 Ti laptop GPU). After the physical installation of the wearable device on the head of the participants, the eye cameras were precisely adjusted. A semi-automated calibration process (using calibration circles appearing on the PC screen) was performed with each participant immediately before the driving test. ID-tag markers (printed on 50 mm × 50 mm hard plastic plates) were placed around the center console of the dashboard and view field of the driver (around the windshield) for optional post-processing reasons.

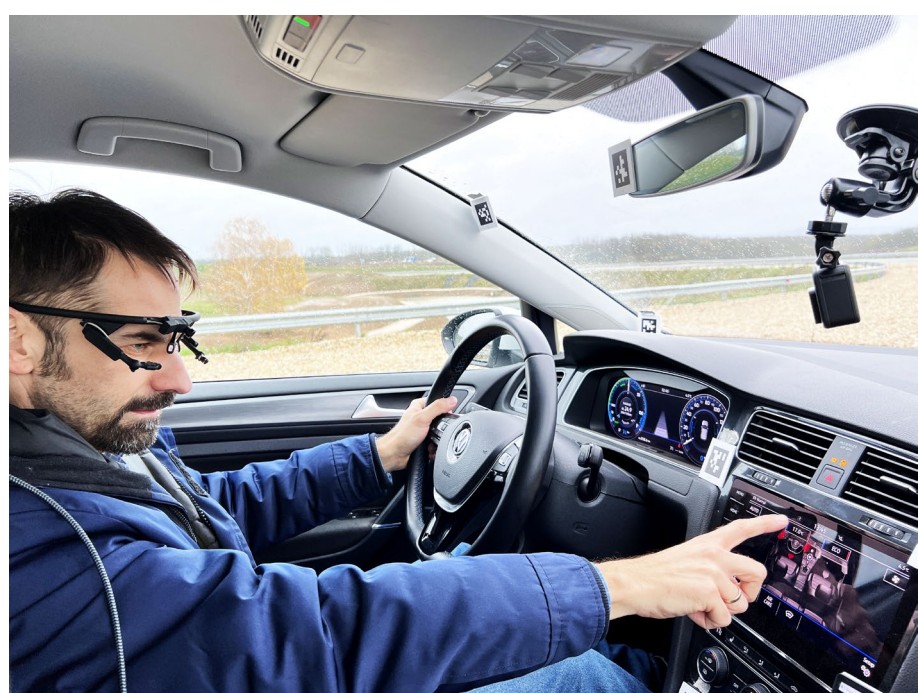

**Figure 2.** Test apparatus: Pupil Core eye-tracking device installed on the head of driver; HD camera in front of the driver; ID tags in the view of driver (around windshield and center console).

B. Procedure

The concept of the pilot test was to measure the differences in visual, manual, and cognitive distraction of a simple, short NDRT using two different types of interfaces. Test circumstances were identical for all participants. The short duration of the tasks assured that the variables (negative environmental effects) were excluded from the study and that the tasks were sufficiently comparable. Drivers encountered similar environmental circumstances as they were driving on a closed track with constant weather conditions. The tasks were conducted on the straight part of the track to eliminate the distracting effect of following the curve of the road and a change of the direction of scattered sunlight coming from the back. The task was to set the internal temperature of the vehicle by using the climate control system; specifically, participants had to increase or decrease the temperature by 2.5 Celsius in five steps, where one step corresponds to a change of 0.5 Celsius. The participants were verbally instructed before the procedure and also shown how to use the actual interfaces in the car before the test. The instructors had a guide in order to always give identical instructions to each participant.

The test vehicle was a Volkswagen e-Golf, production year 2020. It had a traditional, physical climate control panel with buttons and rotating knobs and had the option to use a high definition touch screen to adjust the climate control system (Figure 3). When using physical buttons, the rotating knobs were used to modify the temperature and optionally a "Menu" button could be pushed in order to see all the functions of the system on the screen. The touch screen had multiple functionalities, so the air conditioning function had to be selected from the menu system and the participant had to then push the "Climate" icon on the touch screen. The climate control screen then appeared and the temperature could be set by pushing the dedicated touch screen signs (+ or − "squares" with a red or blue background). First, the physical buttons (Task B) then the touchscreen icons and signs (Task T) were performed to complete the task. The two tasks were completed by the participants in sequence at speeds of 50, 90, and 130 km/h. These are the respective typical speed limits on urban and non-urban roads and motorways in the European Union [55]. The driving speed was defined and limited with the test vehicle's speed limiter system.

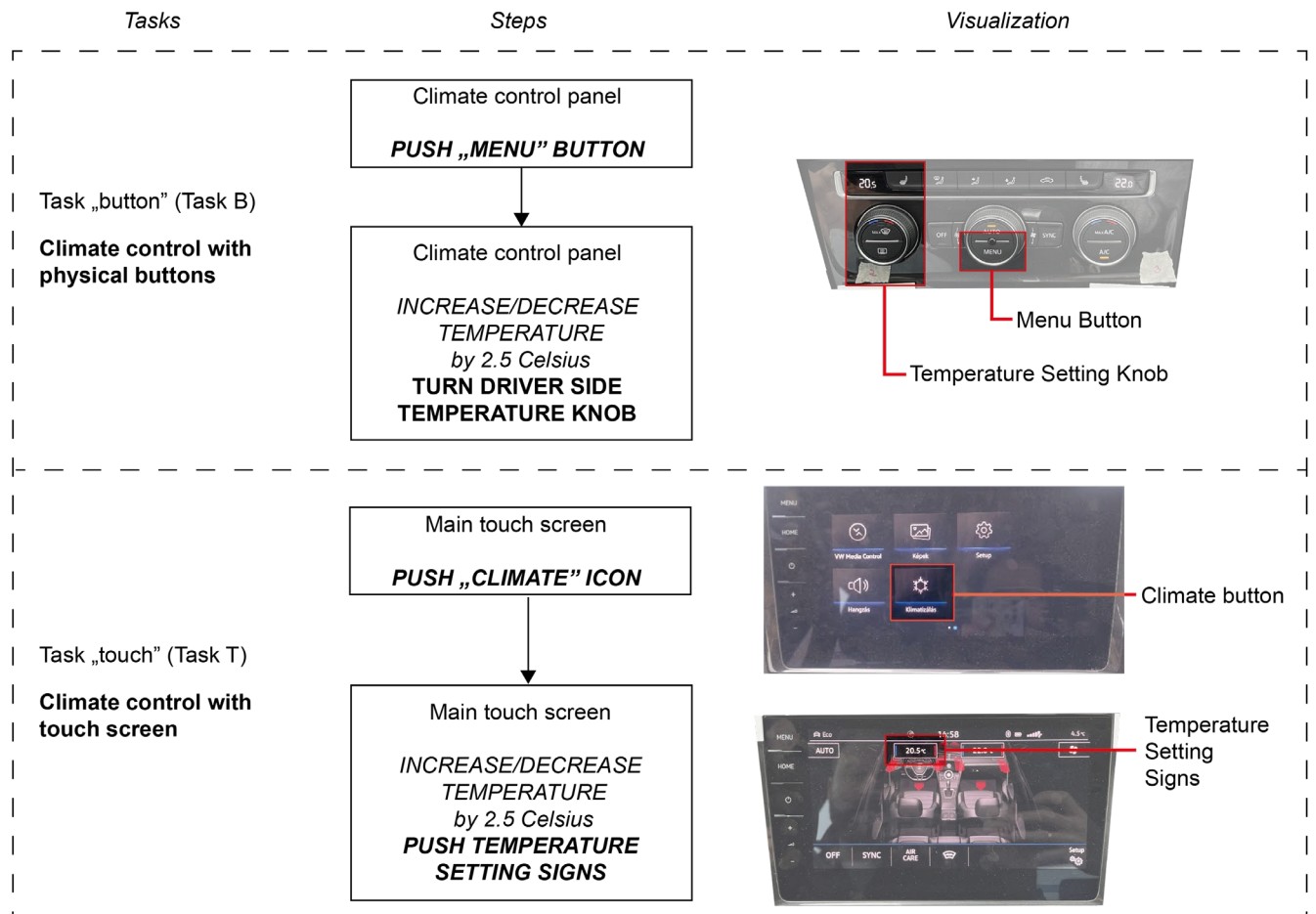

**Figure 3.** Task "button" (Task B) and Task "touch" (Task T) (with control panel icons and signs) performed at different speeds.

Four types of written tests were filled out by all the participants to collect general and experiment-specific objective data and subjective information on cognitive load:

- General questionnaire: demographic data, driving experience, vehicle usage, IVIS "knowledge".
- System Usability Scale (SUS): uses the Likert scale where participants indicate the degree of agreement or disagreement with a statement on a 7-point scale [56].
- Task Load Index (NASA-TLX)–workload assessment: used as a human-centered multidimensional rating based on psychological structure of subjective workload evaluation [57].
- Specific questionnaire: touch vs. button comparison, additional comments.

The general questionnaire was completed before the test; all the others were completed after the test.

Our study covered five objective measured items for which data collection was conducted using the eye-tracking system:

- eyes off the road;
- single hand driving;
- pupillometry;
- saccades.

After that, two typically applied subjective psychological tests were carried out:

- System Usability Scale (SUS);
- Task Load Index (NASA-TLX).

## 3. Results

Post-processing analysis was conducted using the Pupil Player software provided by the manufacturer of Pupil Core wearable eye-tracker device, Pupil Labs. All raw data were exported as a .csv file for further calculations using MS Excel. The measurement results were divided into six different categories according to distraction type. The result categories and data analysis methods are shown in Table 1 in order of presentation.

**Table 1.** Driving test measured items, distraction types, analysis methods and data types.

| Measured Items | Distraction Type | Analysis Method | Data Type |
| --- | --- | --- | --- |
| Total Eyes-Off-Road Time | Visual Distraction | visual, then statistical | spatial, temporal |
| Single Hand Drive | Manual Distraction | visual, then statistical | spatial, temporal |
| Pupillometry | Cognitive Distraction | statistical | dimensional |
| Saccades | Cognitive Distraction | statistical and visual | temporal |

The NDRT requires visual attention and causes visual distraction; this can be measured by gaze tracking and results in Total Eyes-Off-Road Time (TEORT) [58]. TEORT is the time when the driver is not looking at the road, so the Area Of Interest (AOI) is on the IVIS system interface. Eye-tracking captures contain video recordings (world-view camera, Eye0, Eye1) and raw data (time stamp, pupil position, pupil diameter, calculated gaze position on x, y coordinates). World-view camera recordings provide the opportunity to spot manual distractions, in this case Single Hand Drive (SHD) times.

Pupil Player enables the visualization of gaze position projected on the world-view video recording in order to examine participants' eye movements and glance behavior. This is the best and most precise way of identifying and registering place and time of gaze (Figure 4). The GUI enables the researcher to move dynamically on the timeline and to show the eye and world-view camera recordings. Gaze is visualized by a green dot and movements are displayed by green lines. If a fixation is detected then a yellow circle appears and a blue number indicates the number of fixations so far. The lower part of the window displays pupil parameters, blinking, and other important data on the timeline.

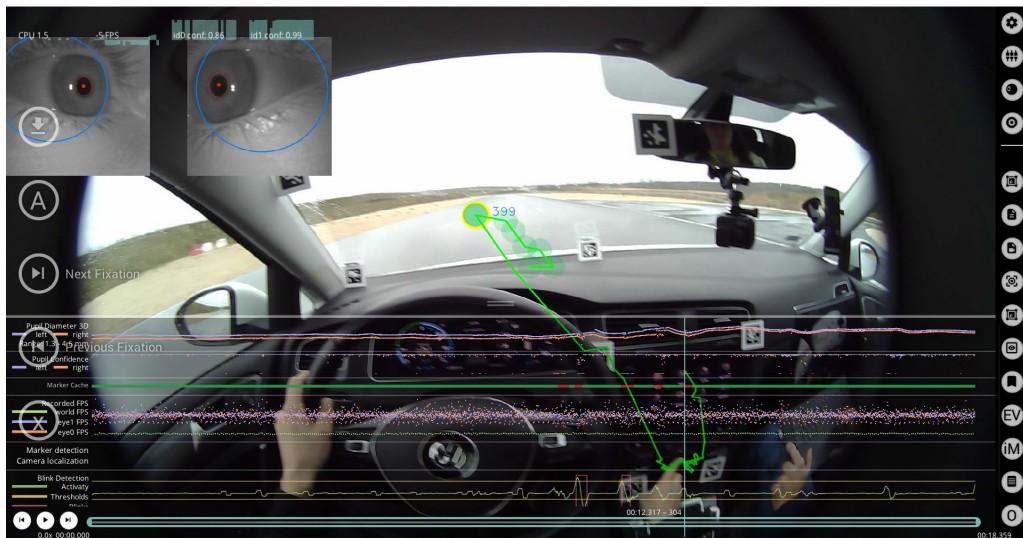

**Figure 4.** Gaze analysis with Pupil Player visualization.

### 3.1. Eyes off the Road

The duration of visual distraction can be determined by analyzing the eye-tracking data; the outcome of this analysis is the TEORT. TEORT values were summarized from the gaze data of all participants; mean, minimum, maximum, and deviation were then calculated (Table 2). TEORT values show that task completion times are in strong negative

correlation with the driving speed ($r_B = -0.9991$; $r_T = -0.9929$.) This means that if the driving speed is higher, the same NDRT is performed faster. The TEORT values also indicate the difference between the two tasks. Performing Task T takes a longer time than Task B at every driving speed. The relatively high level of standard deviation suggests that each participant produced unique task times with significant differences between them.

**Table 2.** TEORT and statistical values.

| Speed (km/h) | Button (B) or Touch (T) | Mean (s) | Max (s) | Min (s) | SD (s) | MD (s) | CI 95% |
|---|---|---|---|---|---|---|---|
| 50 | B | 3.59 | 5.87 | 2.50 | 1.01 | 0.80 | 0.49 |
| 50 | T | 3.76 | 6.29 | 1.93 | 1.04 | 0.74 | 0.51 |
| 90 | B | 3.03 | 5.62 | 1.46 | 1.07 | 0.84 | 0.53 |
| 90 | T | 3.50 | 6.15 | 1.92 | 1.18 | 0.84 | 0.58 |
| 130 | B | 2.55 | 5.03 | 1.42 | 1.03 | 0.78 | 0.51 |
| 130 | T | 3.11 | 5.65 | 1.96 | 1.01 | 0.72 | 0.49 |

In Table 3 and Figure 5, the Total Eyes-Off-Road Distance (TEORD) was calculated as a function of speed from the TEORT. Results show that the task carried out on the touch screen took a longer time at every vehicle speed. As the driving speed increased in both tasks (Task T and Task B), completion time, which is equal to TEORT, decreased, but a higher speed meant longer distances were driven with Lack Of Attention (LOA). The time difference between Task B and Task T increased, meaning that even higher TEORDs emerged. At 130 km/h speed, Task T had a higher distraction level, the time difference was 22.12%, and the difference in distance was 20.34 m.

**Table 3.** Visual distraction and total eyes-off the road distances.

| Speed (km/h) | Button (B) | | Touch (T) | | Difference (Touch to Button) | | |
|---|---|---|---|---|---|---|---|
| | Time (s) | TEORD* (m) | Time (s) | TEORD * (m) | % | TEORD * (m) | Time (s) |
| 50 | 3.59 | 49.84 | 3.76 | 52.18 | 4.68 | 2.33 | 0.17 |
| 90 | 3.03 | 75.73 | 3.50 | 87.49 | 15.53 | 11.76 | 0.47 |
| 130 | 2.55 | 91.93 | 3.11 | 112.27 | 22.12 | **20.34** | 0.56 |

(*) Total Eyes-Off-Road Distance.

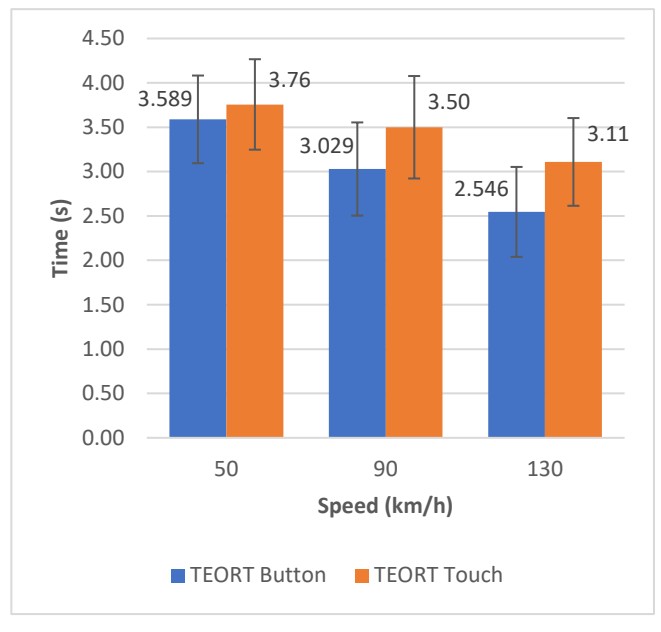

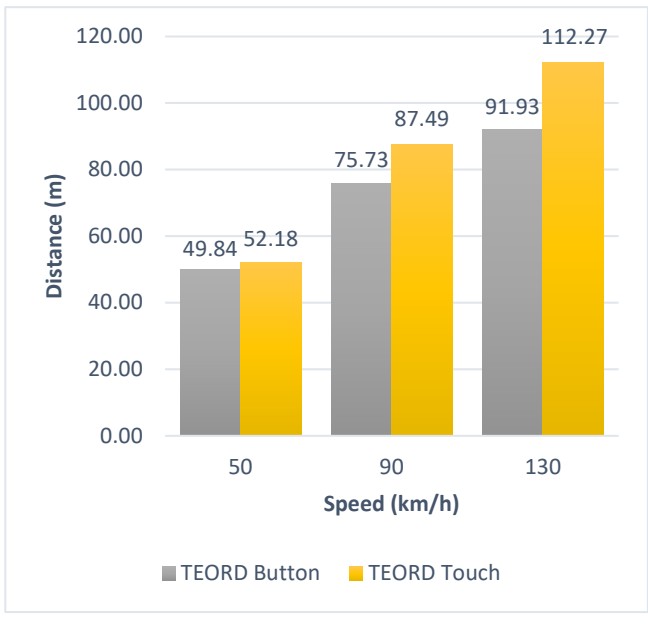

**Figure 5.** (**a**) TEORT and (**b**) TEORD values depending on speed and task type. Error bars denote CI (95%).

A two-way ANOVA was performed to analyze the effect of interface design at different driving speeds. Simple main effects analysis showed that interface design did not have a statistically significant effect on TEORT ($F(1, 90) = 3.44$; $p = 0.0669$), but that driving speed had a statistically significant effect on TEORT ($F(2, 90) = 5.11$; $p = 0.0079$). A two-way ANOVA revealed that there was not a statistically significant interaction between driving speed and interface design ($F(2, 90) = 0.3$, $p = 0.7416$).

### 3.2. Single Hand Drive

Manual distraction can be measured by monitoring the driver's behavior, especially hand movements. The driver needed to release one hand from the steering wheel to complete the task. If an unexpected event occurs (e.g., a wild animal runs across the road) and a necessary maneuver has to be performed with two hands, the risk of an accident is increased because of single-handed operation, or the time lost putting the other (NDRT occupied) hand on the wheel. Single-Hand Drive (SHD) time was detected by visually aided analysis of world-view camera recordings of the eye-tracker device and conducted using Pupil Player. Table 4 shows the average SHD time and statistical calculations.

**Table 4.** SHD and statistical values.

| Speed (km/h) | Button (B) or Touch (T) | Mean (s) | Max (s) | Min (s) | SD (s) | MD (s) | CI 95% |
|---|---|---|---|---|---|---|---|
| 50 | B | 5.35 | 7.71 | 3.65 | 1.30 | 1.08 | 0.64 |
| 50 | T | 5.51 | 7.35 | 4.38 | 0.98 | 0.79 | 0.48 |
| 90 | B | 5.08 | 7.76 | 3.20 | 1.38 | 1.19 | 0.68 |
| 90 | T | 4.96 | 7.95 | 4.00 | 1.15 | 0.79 | 0.56 |
| 130 | B | 4.86 | 7.42 | 3.16 | 1.20 | 0.96 | 0.59 |
| 130 | T | 4.92 | 7.14 | 3.67 | 0.95 | 0.74 | 0.47 |

Mean values in Table 4 demonstrate that SHD times are in strong negative correlation with the driving speed ($r_B = -0.9987$; $r_T = -0.8921$). This means that if the driving speed is higher, the same NDRT is completed faster, and the driver is driving with one hand for a shorter time. The SHD time values also indicate the difference between both tests. Performing Task T is completed slower with one hand than Task B at every driving speed. However, it is also clear that standard deviation and mean deviation values are higher than in the TEORT statistics (Table 2).

In Table 5, the Single Hand Drive Distance (SHDD) was calculated as a function of the speed from the single hand drive time. The results showed that Task T takes no longer than Task B; the difference is within the margin of error. As the driving speed increased, each SHD time decreased, but at higher speeds longer distances were driven with one hand on the steering wheel. At every speed level, quite long distances were driven with one hand and high manual distraction was detected, despite the fact that these tasks were short and simple.

**Table 5.** SHD values and differences.

| Speed (km/h) | Button (B) | | Touch (T) | | Difference (Touch to Button) | | |
|---|---|---|---|---|---|---|---|
| | Time (s) | SHDD * (m) | Time (s) | SHDD (*) (m) | % | SHDD * (m) | Time (s) |
| 50 | 5.35 | 74.32 | 5.51 | 76.54 | 2.99 | 2.22 | 0.16 |
| 90 | 5.08 | 127.10 | 4.96 | 123.95 | −2.47 | −3.14 | −0.13 |
| 130 | 4.86 | 175.54 | 4.92 | 177.74 | 1.25 | 2.20 | 0.06 |

(*) Single-Hand Drive Distance.

Simple main effects analysis showed that interface design did not have a statistically significant effect on SHD (F(1, 90) = 0.12; $p$ = 0.7298), but driving speed had a statistically significant effect on SHD (F(2, 90) = 1.35; $p$ = 0.2644). A two-way ANOVA revealed that there was not a statistically significant interaction between driving speed and interface design (F(2, 90) = 0.01, $p$ = 0.9901).

For a better representation of the results, Type-2 fuzzy sets were introduced [59]. Zadeh proposed that computing with words is "a methodology in which the objects of computation are words and propositions drawn from a natural language." [60]. The initial step is to create an encoder to transform words into Type-2 fuzzy sets. Type-2 fuzzy sets and systems generalize and extend the original Type-1 fuzzy sets; thus, one can manage more uncertainty. Type-1 fuzzy systems work with a fixed membership function, while in Type-2 fuzzy systems, the membership function is fluctuating. A fuzzy set determines how input values are converted into fuzzy variables, in our case the formulation of statement that something represents "Higher Cognitive Load", in which expression the "Higher" and the "Cognitive Load" are both fuzzy measures and sets [61,62].

Type-2 fuzzy sets are capable of handling multiple uncertainties due to the conditions of the test carried out, the relatively small values of parameters, and the relatively high standard deviations.

The fuzzy set "Higher Cognitive Load" (HCL) was constructed as follows:

Let $N$ be the number of test participants.

Let $R_i$ $(i = 1 \dots N)$ be the observed comparison, that is $R_i^{50} = 1$ if the $i^{th}$ participants parameter (potential cognitive load: e.g., time, pupil diameter, saccades) is higher at 50 km/h driving speed, and $R_i^{50} = -1$ when the value is smaller.

The membership function is Type-2 fuzzy:

$$\mu^{50}(HCL) = \alpha \frac{\sum_{i=1}^{N} R_i | R_i = 1}{N} \tag{1}$$

and

$$\alpha = \begin{cases} 1 & if \quad \overline{P}_T + 2\sigma_T \geq \overline{P}_B + 2\sigma_B \\ & and \\ 1 - \dfrac{\overline{P}_B + 2\sigma_B - (\overline{P}_T + 2\sigma_T)}{\overline{P}_B} & otherwise \end{cases} \tag{2}$$

where:

$\overline{P}_T$ and $\overline{P}_B$ are mean values of observed results of Task T and Task B.

$\sigma_T$ and $\sigma_B$ are the SD of the above statistics.

Example:

SHD at 50 km/h

$$\overline{P}_B = 5.35; \ \sigma_B = 1.3$$

$$\overline{P}_T = 5.51 \ ; \ \sigma_T = 0.98$$

$$\overline{P}_B + 2\sigma_B = 5.35 + 2 \times 1.3 = 7.95$$

$$\overline{P}_T + 2\sigma_T = 5.51 + 2 \times 0.98 = 7.47$$

Thus

$$\alpha = 1 - \frac{7.95 - 7.47}{5.35} = 0.91$$

$$\mu^{50}(HCL) = 0.91 \times \frac{11}{16} = 0.63$$

Therefore, 0.63 is the maximum value of an HCL case, because even if the $\overline{P}_B$ parameters are more favorable in each test result, the larger SD ($\sigma_B$) reminds us of the uncertainty of the comparison.

The Type-2 fuzzy method was applied for all TEORT and SHD values shown in Table 6. TEORT values show the intensity of cognitive load with greater reliability; SHD values show uncertainty in the comparison, because of higher SD.

**Table 6.** Type-2 fuzzy analysis for TEORT and SHD.

| Speed (km/h) | TEORT ($\mu$) | SHD ($\mu$) |
|:---:|:---:|:---:|
| 50 | 0.88 | 0.63 |
| 90 | 0.75 | 0.50 |
| 130 | 0.88 | 0.46 |

### 3.3. Pupillometry

Pupil diameter changes in response to light. The constant light conditions required for the study were indirectly ensured by keeping the weather conditions and the intensity of scattered sunlight unchanged during the short time intervals required to perform the tasks. The pupillometry test was also performed in the Pupil Player program and post-processing was conducted using MS Excel. Since the recording of the pupil diameter is continuous, the set of points is also continuous, except where blinking occurs. When the participants are blinking, data are not recorded, or recorded only incorrectly. These data are filterable by using an 80% confidence value. The pupil diameter change measured during Task B and Task T was compared by speed category. The study measured the average, maximum, and minimum pupil diameters during the test. The values used for the comparative analysis were calculated from the difference between the minimum and maximum values and mean deviation.

Figure 6 shows that pupil diameter changed more during the 50 km/h task for five participants during Task B and for 11 participants during Task T. In most cases, the difference was significant (at least 0.5 mm). When performing the 90 km/h task, six out of 16 participants' pupil diameter changed more during Task B than during Task T, while 10 participants' pupil diameter changed more during Task T. In most cases, the difference was significant (at least 0.5 mm). When performing the 130 km/h task, four out of 16 participants' pupil diameters changed to a greater extent during Task B and 12 participants during Task T. In nine cases, the difference was significant (at least 0.5 mm).

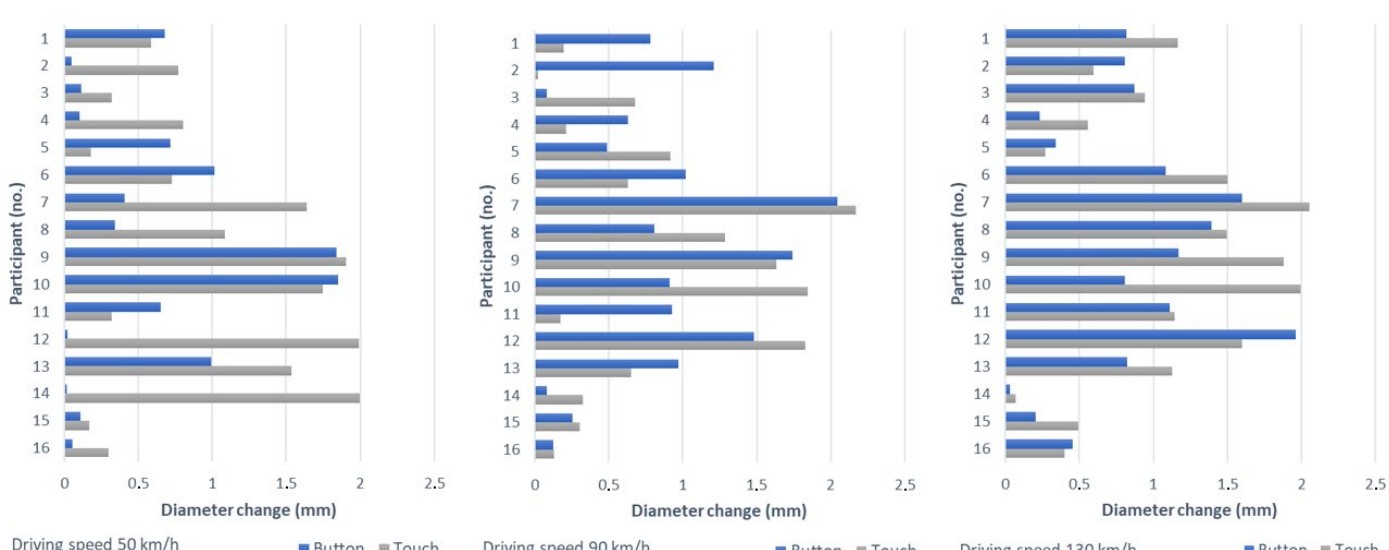

**Figure 6.** Pupil diameter change.

Table 7 illustrates whether cognitive distraction could have occurred while performing the tasks. Cognitive load is positively correlated with pupil diameter change. The change in pupil diameter indicates whether Task B or Task T had a greater effect on cognitive abilities. Assuming that the driver will experience a greater cognitive load while performing Task T, the values are determined as follows:

$$T \geq B \rightarrow 1; T < B \rightarrow -1 \ (red)$$

**Table 7.** The level of cognitive load based on pupil diameter: T ≥ B →1; T < B → −1 (red).

|  | 1 | 2 | 3 | 4 | 5 | 6 | 7 | 8 | 9 | 10 | 11 | 12 | 13 | 14 | 15 | 16 |
|---|---|---|---|---|---|---|---|---|---|---|---|---|---|---|---|---|
| **50** | −1 | 1 | 1 | 1 | −1 | −1 | 1 | 1 | 1 | −1 | −1 | 1 | 1 | 1 | 1 | 1 |
| **90** | −1 | −1 | 1 | −1 | 1 | −1 | 1 | 1 | −1 | 1 | 1 | 1 | −1 | 1 | 1 | 1 |
| **130** | 1 | −1 | 1 | 1 | −1 | 1 | 1 | 1 | 1 | 1 | 1 | −1 | 1 | 1 | 1 | −1 |

From the values in Table 7, it can be observed that cognitive load during Task T is higher at all speeds ($\mu_{50} = 0{,}69; \mu_{90} = 0{,}63; \mu_{130} = 0{,}75$). The results obtained in this study clearly indicate that the use of a touch interface (Task T) while driving imposes a higher cognitive load on the driver, posing a greater risk of accidents. Furthermore, pupillometry testing is a very effective method for analyzing cognitive load, especially when performing relatively short tasks. Further research is needed for tests with higher confidence.

### 3.4. Saccades

The number of saccades and the length of the saccades were analyzed during the execution of the tasks. The saccades were differentiated by the following parameters: $DISP_{max} = 0.31°$; $DUR_{min} = 10$ m; and $DUR_{max} = 30$ m. During the investigation, the length of time required for the driver to perform the task was recorded. The results were compared by speed category, so three types of comparisons were made. In the categories, a participant's results were compared to their own results (and not to the other participants). Thus, the possibility of deviations arising from the physical, mental, and behavioral differences of the participants compared to each other were excluded. For each speed category, three charts were used to compare the number of saccades in the two tasks (Figure 7). The duration of saccades is very low, so tasks of about 4–5 s were a sufficient length for comparison. In evaluation, the number of saccades per unit of time (t = 1 s) was visualized for comparability.

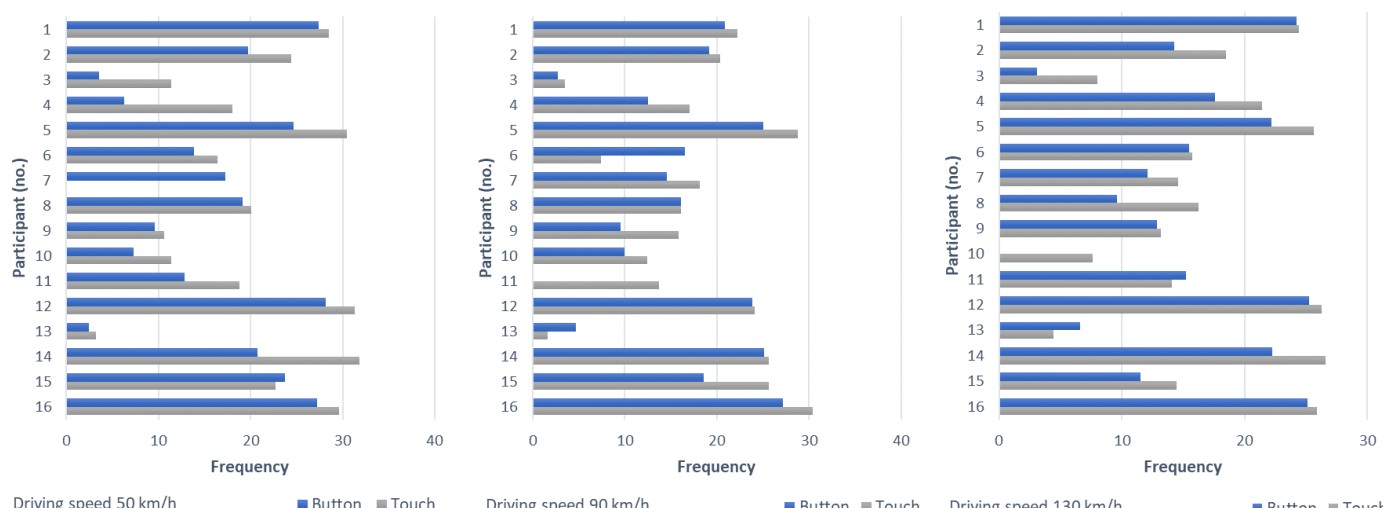

**Figure 7.** Frequency of saccades (t = 1 s).

Figure 7 shows that most of the participants had more saccadic eye movements when performing Task T. The graph shows that, per participant, the individual results are very close, while the results of the individuals relative to each other differ significantly. This also explains the justification for using a comparative evaluation method per participant.

Table 8 illustrates whether cognitive distraction could have occurred while performing the tasks. Cognitive load is positively correlated with the number of saccades. The higher value of saccades indicates whether Task B or Task T had a greater effect on the cognitive abilities. Assuming that the driver will experience a greater cognitive load while performing Task T, the values are determined as follows:

$$T \geq B \rightarrow 1; T < B \rightarrow -1 \ (red)$$

**Table 8.** The level of cognitive load based on saccade frequency: T ≥ B →1; T < B → –1 (red).

|  | 1 | 2 | 3 | 4 | 5 | 6 | 7 | 8 | 9 | 10 | 11 | 12 | 13 | 14 | 15 | 16 |
|---|---|---|---|---|---|---|---|---|---|---|---|---|---|---|---|---|
| **50** | 1 | 1 | 1 | 1 | 1 | 1 | −1 | 1 | 1 | 1 | 1 | 1 | 1 | 1 | 1 | −1 |
| **90** | 1 | 1 | 1 | 1 | 1 | −1 | 1 | 1 | 1 | 1 | 1 | −1 | 1 | 1 | 1 | 1 |
| **130** | 1 | 1 | 1 | 1 | 1 | 1 | 1 | 1 | 1 | 1 | −1 | −1 | 1 | 1 | 1 | 1 |

The values summarized in Table 8 show that cognitive load is highly present at all driving speeds; only 2 exceptions per speed were observed. The results provide evidence $(\mu_{50,90,130} = 0{,}88)$ that the use of a touchscreen interface (Task T) while driving imposes a higher level of cognitive load on the driver, posing a greater risk of accidents. This study of driver distraction demonstrates that the detection of cognitive load by saccade analysis is a very effective method for testing; however. further research is needed.

*3.5. NASA-TLX*

Subjective mental workload and cognitive distraction measurement is required for validating eye-tracking system results. NASA-TLX was chosen for the reasons below [63]:

- applicability for all driving dual-task scenarios;
- reflecting the cognitive component of workload;
- usability for non-expert participants;
- most-used workload questionnaire across academic publications.

The test contains six subscales, which are as follows:

- **Mental Demand:** How much mental and perceptual activity was required (e.g., thinking, deciding, etc.)? Was the task easy or demanding, simple or complex, exacting or forgiving?
- **Physical Demand:** How much physical activity was required (e.g., pushing, pulling, turning, controlling, etc.)? Was the task easy or demanding, slow or brisk, slack or strenuous, restful or laborious?
- **Temporal Demand:** How much time pressure did you feel? Was the pace slow and leisurely or rapid and frantic?
- **Own Performance:** How successful do you think you were in accomplishing the goals of the task? How satisfied were you with your performance in accomplishing these goals?
- **Effort:** How hard did you have to work (mentally and physically) to accomplish your level of performance?
- **Frustration Level:** How insecure, discouraged, irritated, stressed, and annoyed versus secure, gratified, content, relaxed, and complacent did you feel during the task? [57]

Participants scored on a scale 1–21 but without visible numbers on the scale. The 21 gradations were converted into a value from 0–100 in steps of five. The 21 gradations of the scale seem sufficient to compare IVIS tasks with small differences [63]. The mean scores were calculated and are shown in Table 9 and in Figure 8.

**Table 9.** NASA-TLX points and statistical values.

| Speed (km/h) | Button (B) or Touch (T) | Mean (s) | Max (s) | Min (s) | SD (s) | MD (s) | CI 95% |
|---|---|---|---|---|---|---|---|
| 50 | B | 32 | 59 | 11 | 12.3 | 9.31 | 6.03 |
| 50 | T | 46 | 80 | 14 | 19.87 | 16.45 | 9.73 |
| 90 | B | 35 | 59 | 13 | 13.85 | 10.94 | 6.79 |
| 90 | T | 50 | 80 | 23 | 18.47 | 15.13 | 9.05 |
| 130 | B | 43 | 77 | 17 | 17.24 | 13.45 | 8.45 |
| 130 | T | 57 | 88 | 30 | 17.57 | 15.19 | 8.61 |

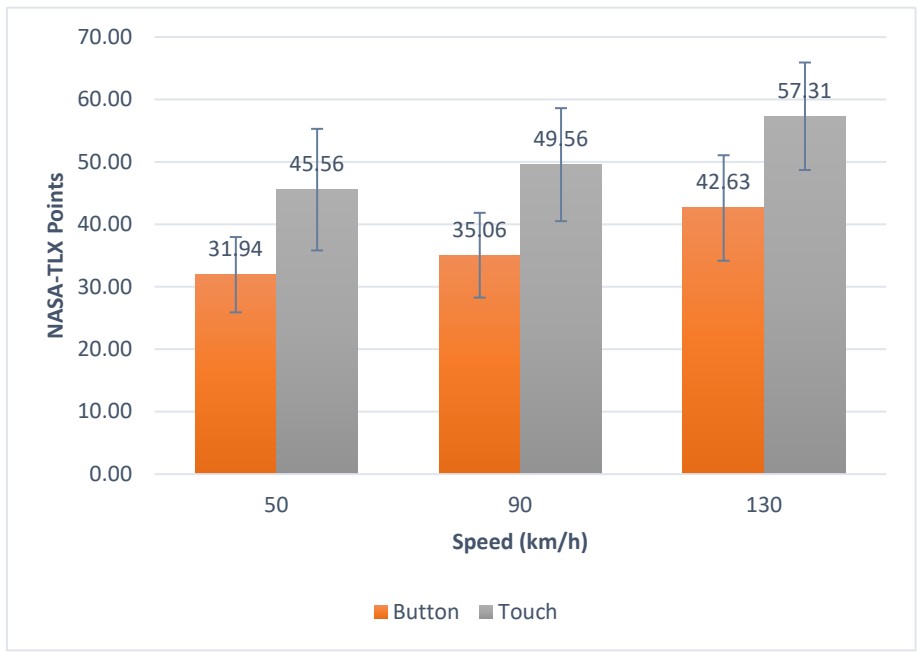

**Figure 8.** NASA-TLX results. Error bars denote CI (95%).

Figure 8 shows that Task T caused higher cognitive load at every driving speed; the difference is 29.9% at 50 km/h, 29.3% at 90 km/h, and 25.6% at 130 km/h. Higher cognitive load means a higher level of driver distraction. Table 9 shows the detailed investigation and analysis of NASA-TLX results for mental demand. Most of the participants (14 of 16, 87.5%) answered that using the touch screen was more mentally taxing; the others felt the same mental demand with each type of UI. The physical demand of the tasks (necessary movements, sequences of movements, coordination of movement) was mostly higher while using the touchscreen (11 of 16, 68.75%) but with the smallest differences; the rest felt it to be identical to the buttons. The subjective perception of the time needed (temporal demand) for the different tasks did not differ significantly. Self-assessment of performance should be treated with caveats because of the potential discrepancy between the perception of error and actual objectively measurable error. Results showed that each task was equally evaluated except with four participants. All participants' performance was "better" at Task B, except for one driver. Effort means how ergonomic and user-friendly the interface is, how well it is integrated into the environment, and how much extra effort is required to use it in order to perform the task properly. All participants found the touch screen less user-friendly except one driver, and for another one it was equal. Frustration

level shows how confident the driver felt in completing the task, or how uncertain, stressed or frustrated the driver may have been during the task. 75% of the participants felt more frustrated with Task T, the others felt the systems equally frustrating. Detailed results NASA-TLX are shown in Table 10.

**Table 10.** NASA-TLX detailed results (Button Task: B; Touch Task: T; Mental Demand: MeD; Physical Demand: PD; Temporal Demand: TD; Own Performance: OP; Effort: E; Frustration Level: FL).

| | 1 | | 2 | | 3 | | 4 | | 5 | | 6 | | 7 | | 8 | | 9 | | 10 | | 11 | | 12 | | 13 | | 14 | | 15 | | 16 | |
|---|---|---|---|---|---|---|---|---|---|---|---|---|---|---|---|---|---|---|---|---|---|---|---|---|---|---|---|---|---|---|---|---|---|
| | B | T | B | T | B | T | B | T | B | T | B | T | B | T | B | T | B | T | B | T | B | T | B | T | B | T | B | T | B | T | B | T |
| MeD | 8 | 13 | 4 | 9 | 2 | 7 | 9 | 14 | 13 | 16 | 13 | 13 | 5 | 6 | 5 | 6 | 4 | 5 | 5 | 9 | 2 | 3 | 10 | 11 | 9 | 12 | 4 | 6 | 9 | 9 | 13 | 13 |
| PD | 9 | 11 | 2 | 4 | 2 | 3 | 7 | 12 | 8 | 8 | 5 | 5 | 2 | 2 | 5 | 6 | 3 | 5 | 5 | 6 | 3 | 4 | 3 | 9 | 2 | 2 | 2 | 2 | 3 | 3 | 5 | 11 |
| TD | 7 | 10 | 1 | 2 | 6 | 13 | 5 | 18 | 10 | 11 | 10 | 10 | 5 | 6 | 7 | 10 | 5 | 5 | 5 | 9 | 4 | 5 | 10 | 10 | 4 | 13 | 7 | 8 | 10 | 10 | 11 | 15 |
| OP | 2 | 5 | 3 | 7 | 4 | 15 | 2 | 4 | 5 | 13 | 4 | 3 | 12 | 3 | 4 | 4 | 3 | 3 | 1 | 2 | 1 | 1 | 5 | 5 | 1 | 1 | 2 | 3 | 3 | 4 | 3 | 12 |
| E | 6 | 19 | 6 | 13 | 9 | 18 | 5 | 13 | 11 | 15 | 12 | 12 | 10 | 12 | 10 | 12 | 5 | 7 | 4 | 10 | 5 | 9 | 10 | 6 | 18 | 19 | 11 | 14 | 11 | 13 | 15 | 16 |
| FL | 5 | 9 | 1 | 2 | 7 | 12 | 7 | 15 | 11 | 15 | 12 | 12 | 10 | 11 | 5 | 8 | 4 | 6 | 4 | 11 | 4 | 8 | 6 | 6 | 2 | 2 | 5 | 7 | 2 | 3 | 12 | 13 |

Table 11 shows that all subscale results are higher while using touch interfaces than using physical buttons. Physical demand and own performance were rated with lower points, meaning that all participants were used to control systems such as a car with IVIS and they did not judge these tasks to be challenging. "Effort" was rated with the highest points because driving speed added a greater mental and physical load.

**Table 11.** NASA-TLX statistical values.

| Points | Mean (Point) | | SD (Point) | | MD (Point) | | CI 95% | |
|---|---|---|---|---|---|---|---|---|
| | B | T | B | T | B | T | B | T |
| MeD | 7.13 | 9.50 | 3.00 | 3.75 | 3.61 | 3.3 | 3.2 | 1.84 |
| PD | 4.06 | 5.81 | 1.71 | 1.77 | 3.14 | 1.4 | 2.6 | 0.87 |
| TD | 6.77 | 9.73 | 2.92 | 2.90 | 3.98 | 2.2 | 2.7 | 1.42 |
| OP | 3.44 | 5.29 | 1.85 | 2.54 | 4.24 | 1.6 | 3.1 | 1.24 |
| E | 9.33 | 12.98 | 3.69 | 3.84 | 3.48 | 3.0 | 3.1 | 1.88 |

Simple main effects analysis showed that interface design did have a statistically significant effect on mental workload according to NASA-TLX ($F_{(1, 90)} = 17.4$, $p = 0.0001$), and showed that driving speed had a statistically significant effect on mental workload ($F_{(2, 90)} = 3.74$; $p = 0.0275$). A two-way ANOVA revealed that there was not a statistically significant interaction between driving speed and interface design according to NASA-TLX ($F_{(2, 90)} = 0.01$; $p = 0.9901$).

*3.6. SUS*

System Usability Scale is a simple ten-item scale that provides a global view of subjective assessments of system usability. The questionnaire contained 10 questions that the participants answered at the end of the experiment, indicating their opinion on a five-point Likert scale. Participants rank each question from one to five based on how much they agree with the statement they read. Five means they completely agreed and one means they completely disagreed. The statements are as follows [56]:

1. I think that I would like to use this system frequently.
2. I found the system unnecessarily complex.
3. I thought the system was easy to use.
4. I think that I would need the support of a technical person to be able to use this system.
5. I found the various functions in this system were well integrated.
6. I thought there was too much inconsistency in this system.
7. I would imagine that most people would learn to use this system very quickly.

8.  I found the system very cumbersome to use.

9.  I felt very confident using the system.

10. I needed to learn a lot of things before I could get going with this system.

The questions, or more precisely statements, cover many aspects of the system's usability. By alternating between positive and negative items, the respondent must read each statement and think about whether they agree or disagree. SUS scores were calculated as follows:

- Item 1, 2, 3, 5, 7, 9 (odd) scores were summarized, then five was subtracted from total to obtain (X).

- Item 2, 4, 6, 8, 10 (even) scores were summarized, then this total was subtracted from 25 to obtain (Y).

- Then (X + Y) × 2.5 resulted in the SUS score.

The total possible score was 100 and each statement had a weight of 10 points. The average SUS score was 68. This means that a score of 68 is at the 50th percentile. The general guideline on the interpretation of the SUS score is shown in Table 12.

**Table 12.** SUS score rating guideline.

| SUS Score | Grade | Adjective Rating |
|-----------|-------|------------------|
| >80.3 | A | Excellent |
| 68–80.3 | B | Good |
| 68 | C | Okay |
| 51–68 | D | Poor |
| <51 | F | Awful |

As seen in Table 13, the mean of the SUS score is 53.6% higher when using physical buttons when compared to a touchscreen. The button interface was rated as grade "A", Excellent, as shown in Table 12. In contrast, the touchscreen interface received a "D", Poor, rating from the participants. Also, minimum and maximum values show the same difference. This means that all participants evaluated similarly, so the typical difference in the usability of the two systems were the same.

**Table 13.** SUS scores and statistical results at all driving speeds.

| Button (B) or Touch (T) | Mean (Score) | Max (Score) | Min (Score) | SD (Score) | MD (Score) | CI 95% |
|-------------------------|--------------|-------------|-------------|------------|------------|--------|
| B | 81.4 | 100.0 | 50.0 | 14.78 | 12.38 | 7.24 |
| T | 53.0 | 74.2 | 27.5 | 15.10 | 12.37 | 7.40 |

Simple main effects analysis showed that interface design did have a statistically significant effect on system usability according to SUS ($F_{(1, 30)} = 26.42$, $p < 0.0001$).

*3.7. Summary of Results*

In Table 14, the results of the pilot study summarized. Our hypothesis was that Task T causes higher visual distraction, manual distractions, and cognitive load (cognitive distraction).

**Table 14.** Summary of comparison results.

| Measured Items | Finding | Difference (Task B to Task T) | Results |
|----------------|---------|-------------------------------|---------|

| | | | |
|---|---|---|---|
| TEORT | Mean values are higher in Task T | 4.68% at 50 km/h, 15.53% at 90 km/h, and 22.12% at 130 km/h | Only driving speed has significant effect; Task T has higher cognitive load in comparison with Task B |
| SHD | No appreciable difference observed | 2.99% at 50 km/h, −2.47% at 90 km/h, and 1.25% at 130 km/h | Only driving speed has significant effect; Task T has higher cognitive load, but with uncertainty |
| Pupillometry | Higher cognitive load detected | 69% at 50 km/h, 63% at 90 km/h, and 75% at 130 km/h | Pupil diameter change can indicate cognitive load |
| Saccades | Higher cognitive load detected | 87.5% at all speeds | Frequency of saccades can indicate cognitive load |
| NASA-TLX | All points of subscales show higher load in Task T | 29.9% at 50 km/h, 29.3% at 90 km/h, and 25.6% at 130 km/h | Significant effect; Cognitive load is higher in Task T; higher driving speed means higher cognitive load |
| SUS | Task T has higher system usability difficulty and complexity | 53.6% | Significant effect; Task B got Grade "A", Task T got Grade "D" |

## 4. Discussion

The goal of our study was to investigate driver distraction and cognitive load in a naturalistic driving test conducted on a closed track comparing different UIs by using a wearable eye-tracking device and psychological questionnaires.

Previous studies showed that IVIS interactions performed while driving causes higher cognitive demand and more distraction [3,9,10]. The level and type of distraction depends on the NDRT and UI. A wide variety of multimodal in-car interactions were investigated including physical buttons, dials, touch screens with different GUI layout, and also speech and gesture control [13–15,22–24]. Besides other examination techniques, eye-tracking and glance behavior analysis were capable of monitoring driver distraction [30]. It was argued earlier that, under certain circumstances, saccades and pupillometry analyses were suitable for the detection of cognitive load and change in average pupil diameter indicated the difficulty of the task [36–38,40,42,46].

The conditions and technique of the comparative pilot study were designed to build on the findings of previous research while still producing new results. Under identical driving conditions, the test participants used two types of user interfaces (physical button layout and touch screen) to perform a short NDRT (adjusting internal temperature). In order to minimize variables in the current study (e.g., lighting conditions), short tasks were completed under constant weather conditions. The interfaces were part of a series production car, which ensured real User Experience (UX) and high-quality ergonomic design, so results could differ from other studies using retrofitted interfaces. In order to examine cognitive factors, distraction was detected by using eye-tracking data acquisition. The pilot study examined visual, manual, and cognitive distractions, that were monitored, analyzed, and tested using different statistical methods. In addition to eye tracking data, the most common psychological tests were used. The subjective NASA-TLX and SUS tests were carried out, and coincided with the results measured by the eye-tracking system. This means that the participants' behavior and emotions correlated in the comparison study.

There were limitations and restrictive conditions of the test. First, the pilot test was carried out with the participation of 16 individuals. The chosen NDRTs were simple and short so that data collection and analysis could be performed using a broader spectrum of

methods, and the conditions of the study were more controllable. The test vehicle was a mass production passenger car and the IVIS used can typically be found in several types of vehicles. This was an advantage because most of the participants were used to the system, so general ergonomics was adequate; however, this also limited the functionality and layout variations used for the test.

A Type-2 Fuzzy logic system was introduced to evaluate the results. Type-2 Fuzzy logic is suitable in projects when there are a lot of imprecisions and uncertainty of data. The small number of participants and relatively high SDs explain the need to use "Higher Cognitive Load" as a fuzzy set. This allows us to compare low element count, naturalistic driving study data, and display the subtle differences caused by the human factor. In our comparison study, more sophisticated test result showed up in addition the conventional statistical tests.

The results of the comparative study are summarized as follows:

1.  Visual distraction times and eyes-off-road distances clearly demonstrate the difference between the two types of interface design and high traffic safety risk of using IVIS in moving vehicle.

2.  In all three driving speeds (50, 80, 130 km/h), the touch screen-based interface caused higher visual distraction (4.68–22.12%), resulting in a maximum of 20.34 m eyes-off-road distance difference.

3.  Manual distraction times and single-hand drive distances showed a small degree of difference, but in some cases physical buttons could be used without visual attention. This is safer when performing a NDRT.

4.  Clear differences between the examined different UI designs showed a higher distraction level when using the touchscreen.

5.  Pupil diameter and saccades presented comparable values for showing higher cognitive load in controlled test circumstances and in the comparative study.

6.  NASA-TLX and SUS tests coincided with the results measured by the eye-tracking system.

7.  These results can explain the advantage of physical buttons and turning knobs. Physical buttons can be operated by the driver's arm, hand, and fingers using "muscle memory" and complex haptic and tactile feedback without the need to focus the gaze.

More research is needed to dissociate the effects of light and cognitive load on pupil diameter [64]; this is not only for comparison, as absolute measurement values are needed as well. Our pilot investigation needs refinement and further studies need to be conducted on different types of road vehicles. These could be applicable to public road transportation and rail transportation.

## 5. Conclusions

The comparative naturalistic driving study was conducted as a pilot to investigate conventional (tactile) and touch interfaces used in mass-production road vehicles. Visual, manual, and cognitive distraction were examined using an eye-tracking system and psychological questionnaires. Besides traditional statistical tests, Type-2 fuzzy sets were added for better representation of results. Results showed that the operation of touchscreen interface integrated to IVIS causes more driver distraction even when performing simple NDRTs. Further research is needed on the applicability of the eye movement and pupilometry-based cognitive load test adapted to driving.

It is important to note that the suggested method can be used to examine additional problems of human-computer interaction (HCI). This will increase the safety of transportation systems by understanding human factors, allowing driver behavior to be analyzed and errors to be detected.

**Author Contributions:** Conceptualization, V.N., G.K., and P.F.; methodology, V.N., G.K., and P.F.; software, V.N.; validation, V.N., G.K., and P.F.; formal analysis, V.N., G.K., and P.F.; investigation, V.N., G.K., P.F., D.K., M.S., S.S. and S.F.; resources, V.N.; data curation, V.N.; writing—original draft preparation, V.N., G.K., P.F., D.K., M.S., S.S. and S.F.; writing—review and editing, V.N., G.K., P.F., D.K., M.S., S.S. and S.F.; visualization, V.N.; supervision, G.K., P.F., D.K., M.S., S.S. and S.F.; project administration, V.N., G.K., and P.F; funding acquisition, V.N., G.K., and P.F. All authors have read and agreed to the published version of the manuscript.

**Funding:** This research did not receive any specific grant from any funding agencies.

**Data Availability Statement:** All data of this research were presented in the article.

**Acknowledgments:** The authors wish to acknowledge the support received from the Vehicle Industry Research Centre and Széchenyi István University, Győr. The research presented in this paper was supported by the ELKH-SZE Research Group for Cognitive Mapping of Decision Support Systems. The research was carried out as a part of the Cooperative Doctoral Program supported by the National Research, Development and Innovation Office and Ministry of Culture and Innovation.

**Conflicts of Interest:** The authors declare no conflict of interest.

## Abbreviations

| | |
|---|---|
| ADAS | Advanced Driving Assist Systems |
| AOI | Area Of Interest |
| B | Button |
| DDA | Driver Diverted Attention |
| E | Effort |
| FL | Frustration Level |
| GUI | Graphical User Interface |
| HCI | Human-Computer Interaction |
| HCL | Higher Cognitive Load |
| HMI | Human-Machine Interfaces |
| IVIS | In-Vehicle Information Systems |
| LOA | Lack Of Attention |
| MD | Mean Deviation |
| MeD | Mental Demand |
| NASA-TLX | NASA Task Load Index |
| NDRT | Non-driving Related Task |
| NDRTs | Non-driving Related Tasks |
| NHTSA | National Highway Traffic Safety Administration |
| OffCT | Offset Confidence Threshold |
| OnCT | Onset Confidence Threshold |
| OP | Own Performance |
| PD | Physical Demand |
| POR | Point Of Regard |
| SD | Standard Deviation |
| SHD | Single Hand Drive |
| SHDD | Single Hand Drive Distance |
| SUS | System Usability Scale |
| T | Touch |
| Task B | Task "button" |
| Task T | Task "touch" |
| TD | Temporal Demand |
| TEORD | Total Eyes-Off-Road Distance |
| TEORT | Total Eyes-Off-Road Time |
| UI | User Interface |

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
