# Peer review of "Testing Road Vehicle User Interfaces Concerning the Driver’s Cognitive Load"

_infrastructures, doi:10.3390/infrastructures8030049_

Round 1

Reviewer 1 Report

The authors study the differences between in-production traditional style knob in-car UI and in-car touchscreen UI for a climate control task in a track study with 16 participants, and compare the UIs with a number of different subjective and objective measures, including questionnaires and eye-tracking measures. The topic of the research is highly important and there is a clear need for better testing methods for in-car UIs than the existing standards and suggested means.

However, the presented study does not offer a real improvement to SotA and there are such major deficiencies in the manuscript that it should not be published in its current form. As a summary, the relevant research has not been reviewed appropriately in the Introduction and the findings are not discussed against this in the end. There are a number of critical methodical details missing. There are no statistical tests and effect sizes reported. There are a number of uncontrolled confounding factors possibly affecting the cognitive load metrics. These haven’t been acknowledged or taken into account in the analysis. The findings are quite obvious if you are familiar with the related research. I am not sure if a revision would help in getting the manuscript into a level it could be published but I wish the best for the authors in their future endeavours and I hope they’ll find my feedback useful. This seemed to be a pilot study, like they themselves write.

Detailed comments by section:

Abstract
- In-Vehicle Information Systems (IVIS)
- “Actual attention” (even visual) cannot be detected with eye-tracker. There can be a mismatch between where the driver is looking at and where the driver’s visual attention is. This should be acknowledged.

Introduction
- What is “UX factor”, and how it is “fundamentally unchanged”? In-car UIs have developed a lot within the last 10 years. Please clarify.
- Human-Computer Interaction
- What is “infocommunications”? Please define.
- In my opinion the authors put too much emphasis on automated driving in the Introduction. This type of testing is maybe even more important for manual driving. Moreover, it will be a very long time until all cars are driving autonomously on the roads (if ever).
- As you study touchscreen vs. more traditional knob/rotor in-car interfaces, you should review all the existing studies on this topic (there is a lot of this literature available, you should not ignore it!).

I. Human Machine Interface
- What is “safety awareness level”?

IV. Cognitive load
- “Secondary knowledge – e.g., driving a vehicle – is first processed by a working memory…” This is a really odd and unconventional description of driving a vehicle, from a psychological point-of-view. Please remove or clarify. I don’t understand the whole point of the first paragraph, consider removing it.

V. Observing cognitive load
- “Due to continuous, low-speed eye movements during fixation…” I believe these are very fast but small movements.
- “driving test simulations” Do you mean driving simulations or simulators? Driving tests are taken in a driving school.
- “it was shown that average fixation duration is negatively correlated with cognitive load.” Naturally, there can be a high number of other variables that can also affect fixation durations. The same applies for pupil size, saccades and blinks. These should be controlled for, if you want to refer one’s cognitive load from fixation, pupil size, saccadic, or blink data.
- Is there really such a term in psychology as “muscle memory”? Maybe consider using procedural memory instead, or if it is a different thing, then please define it.

VII. The present study
- “Our focus is on precisely monitoring and identifying visual, manual and cognitive distractions with an eye-tracking system and psychological questionnaires.” At this point, I am highly doubtful if you are able to precisely monitor and identify these distractions with your methods but it is a nice goal.
- Typically, you cannot make conclusions of risk in real traffic from a controlled study, even if done on a test track. Maybe you should focus on detecting distractions in your claims.

2. Experiment
- Please report SD and range of participants’ age.
- What is “up-to-date IVIS”? Please define the criteria.
- B. Methodology ==> Procedure (Methodology refers to a study of methods.)
- Figure 3: Please clarify in more detail all the necessary steps to complete the two tasks.
- There are standardized metrics for assessing visual load by in-car tasks, e.g., glance durations (e.g., SAEJ2396, NHTSA 2013 guidelines, AAM guidelines). Why didn’t you measure and report these? This would have made your results comparable with the wealth of studies available comparing in the HF literature. Science should be a systematic endeavour building upon others’ findings and existing knowledge.
- More details are needed of the method and procedure, how were the drivers instructed for the tasks? Did you counterbalance the orders of the UIs? How was the exact procedure of a single experiment from a participant’s point-of-view? Were the drivers compensated? Did they write informed consent? Did you have an ethical review for the study? Why this straight test road? Why these measures? etc.

3. Results and discussion
- What does “visual analysis” refer to? How was this done? By a single or many researchers? If by many, what was the IRR?
- “TEORT values show that task completion times are in negative correlation with the driving speed.”  Pearson’s R?
- “considering the human factor” What is this?
- Please report statistical tests with effect sizes when comparing the two conditions. You cannot claim about differences without showing that these were statistically significant, and how strong the effects were.
- “higher speed means longer distances driven with lack of attention (LOA) and without gaze on the road” This is obvious, the same glance time means longer distances with higher speeds.
- Overall, many researcher on the field agree that TEORT does not equal “visual distraction”. What matters for safety, is how the drivers are able to time-share their visual attention between driving and the NDRTs. Se, e.g., Kircher, K., & Ahlstrom, C. (2017). Minimum required attention: a human-centered approach to driver inattention. Human Factors, 59(3), 471-484.
- Figure 5: Rephrase the title, “visual distraction” is misleading, as you only report TEORTs (use TEORT in the title, as that is what you illustrate).
- Figure 5: Please add error bars (e.g., 95% CI).
- Anyone can drive a car on a straight road with a single hand. Therefore I don’t think that the hands-off-wheel metric tells anything meaningful about the distraction potential of the tasks on real roads.
- What is “Type-2 fuzzy sets” and why would you need that here? Just report the statistical test results. This just unnecessarily complicates your results.
- Why would you need all that fixation data? What are their relationship to visual or cognitive distraction? Please clarify.
- There is also a lot of missing fixation data. I recommend to just delete these unnecessary details and graphs from the manuscript.
- Were the lighting conditions controlled for in your pupil data analysis? I guess not, as the experiment was done outside. The variability in the pupil diameters could be to the variability in lighting and this data is therefore highly unreliable.
- Overall, all the suggested and reported metrics for the measurement of cognitive load seem highly unreliable. You should correlate these metrics with NASA-TLX, to analyze their face validity.
- In general, the reporting of the results is poor as there are no statistical tests. For each DV, you should also report SDs and 95% CIs.

Conclusions
- “The study has measured and recorded visual, manual and cognitive distraction, …” You should not claim this, as the relationships of these concepts and your DVs are highly questionable.
- It is interesting that there seems to be such clear differences in the favor of the traditional UI but this is something that can be expected if you are familiar with the existing research literature on the topic, that is overlooked in the manuscript. In a proper Discussion of a scientific article, one should reflect one’s findings against the existing knowledge and state of the art research.

Please proof-read the text by a native English-speaker, who is familiar with the domain.

Author Response

See the attached PDF file in which there is a comparison section (all changes tracked).

Reviewer 2 Report

There have been many works done on driver distraction, an important topic in the field of traffic safety, and this paper is an interesting one.

1-      The abstract did not clarify the location of the case study.

2-      There was no clear indication of the final result of the paper in the abstract.

3-      In the introduction, clarify the SAE level.

4-      I would appreciate it if you could provide a table with all the abbreviations in it.

5-      It would be helpful to summarize the materials presented in Section 1.

6-      Among the road classifications, I am unfamiliar with non-urban and motorways. Could you please explain them based on the AASHTO Green Book?

7-      Using two-way ANOVA, I suggest proving what was presented in Figure 5.

8-      “In order to better represent the results, type-2 fuzzy sets were introduced.” I didn't understand the rationale behind using fuzzy sets.

9-      Before concluding, please summarize all the research's accomplishments.

10-   In Section 1, please present all the research questions.

Author Response

(The authors gave the same response as above.)

Reviewer 3 Report

The article is prepared at a high scientific and formal level. I recommend publishing it in its current form.

Author Response

(The authors gave the same response as above.)

Round 2

Reviewer 1 Report

Thank you for the responses to my earlier comments. There is still a Discussion missing that would reflect the findings with existing literature. If the journal is willing to publish a pilot study, then they should do it.

Author Response

See attached PDF file.
